# Constraining Non-Negative Matrix Factorization to Improve Signature Learning

## Abstract

Collaborative filtering approaches are fundamental for learning meaningful low-dimensional representations when only association data is available. Among these methods, Non-negative Matrix Factorization (NMF) has gained prominence due to its capability to yield interpretable and meaningful low-dimensional representations. However, one significant challenge for NMF is the vast number of solutions for the same problem instance, making the selection of high-quality signatures a complex task. In response to this challenge, our work introduces a novel approach, Self-Matrix Factorization (SMF), which leverages NMF by incorporating constraints that preserve the relationships inherent in the original data. This is achieved by drawing inspiration from a distinct family of matrix decomposition methods, known as Self-Expressive Models (SEM). In our experimental analyses, conducted on two diverse benchmark datasets, our findings present a compelling narrative. SMF consistently delivers competitive or even superior performance when compared to NMF in predictive tasks. However, what truly sets SMF apart, as validated by our empirical results, is its remarkable ability to consistently generate significantly more meaningful object representations.

## 1 Introduction

Learning low-dimensional representations that capture key features of objects (also called signatures) is an important strategy in different tasks such as recommender systems, link prediction on graphs, and clustering (Zeng et al., 2017; Sturluson et al., 2021; Kim & Park, 2007). Collaborative filtering, a technique that relies on object association patterns for uncovering new associations, is essential for learning meaningful signatures when no prior supplementary knowledge about the objects is available. Non-negative Matrix Factorization (NMF), has been adopted as a valuable tool in this context (Lee & Seung, 1999). It demonstrates a strong ability to learn interpretable signatures from complex data, providing valuable insights into the latent processes behind the associations (Lee & Seung, 1999). However, one significant challenge is that NMF yields an infinite number of solutions for the same problem instance, rendering the selection of high-quality signatures a demanding task.

Self-Expressive Models (SEMs) are matrix decomposition methods that have been successfully employed in the context of collaborative filtering, and top-$N$ recommender systems Ning & Karypis, 2011. These models operate under the fundamental premise that a data point can be effectively reconstructed using only a limited set of other data points belonging to the same underlying subspace. This premise closely aligns with the essence of collaborative filtering, as SEM has the capability to express each object as a linear combination of a select few that share significant similarities. Although SEM does not directly learn signatures, it does establish relationships among pertinent objects, offering valuable insights for a range of applications (Elhamifar & Vidal, 2013).

In this work, we propose a novel approach for learning representations, or signatures, for objects, namely Self-Matrix Decomposition (SMF), that constrains the space of NMF solutions by incorporating strategies from SEM. Our fundamental idea revolves around the learning of signatures that align with the inherent similarity relationships derived directly from the association matrix. By imposing the reconstruction to use only data points that lie in the same subspace, we are effectively constraining the number of possible solutions for the learned representations. In our comprehensive analysis across two distinct benchmark datasets, we show that SMF consistently outperforms NMF in learning more meaningful object representations by effectively clustering objects into coherent

groups in the low-dimensional space. This can be attributed to the fact that SMF operates within a more confined solution space. These findings highlight the robustness of SMF in capturing intrinsic attributes and meaningful relationships among objects. In addition, we applied SMF for the problem of matrix completion, in which we want to predict missing measurements of a given association matrix. We expect that the more meaningful the signatures are, the better the predictions get. Indeed, results on both datasets suggest that SMF obtains comparable or better predictions than NMF and SEM.

## 2 RELATED WORK

### 2.1 NON-NEGATIVE MATRIX FACTORIZATION

NMF was introduced as a technique to learn part-based representations (Lee & Seung, 1999). This method decomposes a low-rank non-negative data matrix $X \in \mathbb{R}^{n \times m}$ into the product of two non-negative matrices $W \in \mathbb{R}^{n \times k}$ and $H \in \mathbb{R}^{k \times m}$ such that $X \simeq WH$ with $W, H \geq 0$ and $k << \min(n, m)$. One can learn $W$ and $H$ by minimizing the following loss:

$$\min_{W,H} \mathcal{L}_{\text{NMF}}(W, H) = \frac{1}{2}\|X - WH\|_F^2$$
$$\text{subject to } W, H \geq 0. \tag{1}$$

The process of learning $W$ and $H$ imposes a constraint on the inner product of $W_{i,:}$ and $H_{:,j}$, driving it to approximate the corresponding element $X_{ij}$. This constraint naturally encourages any pair of vectors $W_{i,:}$ and $H_{:,j}$ to exhibit a significant overlap in their high-valued components if the objects $i$ and $j$ share an observed association in $X$. This observation suggests that each component within the lower-dimensional space might encode latent processes that underlie the interaction patterns present in the data. In essence, when objects $i$ and $j$ interact, it is indicative of their participation in the same underlying processes, as evidenced by the elevated values of $W_{i,:}$ and $H_{:,j}$ in the corresponding components.

### 2.2 SELF-EXPRESSIVE METHODS

SEM is an approach that was based on the sparse subspace clustering algorithm (Elhamifar & Vidal, 2013). The main assumption underlying SEM is that a data point can be efficiently reconstructed using only a small subset of complete points belonging to the same subspace. In other terms, each row of the data matrix $X$ can be expressed as a sparse representation of the other rows, denoted as $X \simeq MX$, where $M \in \mathbb{R}^{n \times n}$ represents the self-representation coefficient matrix. The fundamental idea is that one should reconstruct each row by assigning a larger coefficient to the rows that are more similar to the row to reconstruct. Following the work by Ning & Karypis (2011), to learn a sparse matrix $M$, SEM minimizes the following loss:

$$\min_{M} \mathcal{L}_{\text{SEM}}(M) = \frac{1}{2}\|X - MX\|_F^2 + \frac{\beta}{2}\|M\|_F^2 + \lambda\|M\|_1$$
$$\text{subject to } M \geq 0 \text{ and } \text{diag}(M) = 0. \tag{2}$$

The non-negativity constraint is applied to $M$ to capture positive aggregations over the items. Additionally, the constraint $\text{diag}(M) = 0$ is imposed to eliminate the trivial solution of representing each data point as a combination of itself. To induce sparsity in $M$, SEM applies elastic net regularization to $M$. This sparsity constraint effectively sets the coefficients in the coefficient matrix $M_{i,:}$ to approximately 0 for data points that have a marginal role in reconstructing $\hat{X}_{i,:}$. This exploits the inherent structure within the data, as it assigns higher coefficients to data points closely aligned within the same subspace as $X_{i,:}$. Importantly, SEM possesses the unique property of encoding the subspaces present in the data $X$ directly into the coefficient matrix $M$.

The specific approach employed by SEM for reconstructing the data matrix $X$ offers insight into its ability to explain newly predicted links. It is widely acknowledged that data points of the same

class tend to cluster together in distinct structures within the feature space (Gu & Zhou, 2009). By reconstructing a data point using other data points within the same subspace, SEM essentially leverages information from objects belonging to the same class.

# 3 SELF-MATRIX FACTORIZATION

Here we present Self-Matrix Factorization (SMF), a novel constrained Non-Negative Matrix Factorization approach for signature learning and matrix completion. Figure 1a illustrates the main idea behind our approach. SMF learns two non-negative matrices $W \in \mathbb{R}^{n \times k}$ and $H \in \mathbb{R}^{k \times m}$, each containing distinct object signatures, such that their product approximates an interaction data matrix, $X \in \mathbb{R}^{n \times m}$ . In other words,

$$X \simeq WH. \tag{3}$$

This is also the model proposed by NMF, the difference resides in the learning, where we constrain the signatures in $W$ to encode sub-structures information from the association data itself. Figure 1b illustrates black dots representing positions of the rows of $X$ in the space, note how the rows are localized in three different subspaces: two lines and a plane. The blue dots correspond to the positions of three highlighted rows ($i$, $p$, and $q$). SMF constrains the learning of $W$ to be aware of object relationships established by the subspaces. Thus, in the illustrated example, object $i$ should be closer to object $p$ than to object $q$ in the signatures space, mimicking their behavior in the high-dimensional space. Figure 1d shows how this information can be preserved by enforcing the dot product of signatures whose objects belong to the same sub-space in $X$ to be higher than the dot product of the signatures of objects localized in different subspaces.

We propose the following loss function for learning a model with the desired properties:

$$\min_{W,H} \mathcal{L}_{\text{SMF}}(W, H) = \frac{1}{2}\|X - WH\|_F^2 + \frac{\lambda_{se}}{2}\|X - [T \circ (WW')]X\|_F^2$$
$$+ \lambda_1\|W\|_1 + \lambda_1\|H\|_1 + \frac{\lambda_2}{2}\|W\|_F^2 + \frac{\lambda_2}{2}\|H\|_F^2 \tag{4}$$
$$\text{subject to } W, H \geq 0.$$

The first term of Equation 4 is the Euclidean distance between the non-negative matrix $X \in \mathbb{R}^{n \times m}$ and the product of two non-negative matrices $W$ and $H$. Minimizing this distance results in projecting high-dimensional data into a low-dimensional representation. The second term in Equation 4 is designed to preserve the subspaces information, similarly to SEM. These subspaces can be preserved in the lower dimensional representation $W$ by enforcing a high inner product between a pair of object signatures that belong to the same subspace in $X$. To achieve this, we introduce a matrix $T \circ (WW')$ that acts as the coefficient matrix in Equation 2. The matrix $T \in \mathbb{R}^{n \times n}$ is predominantly populated with ones, except for the diagonal where elements are deliberately set to zero. This ensures that the SEM term consistently maintains a coefficient matrix with a zero diagonal. Additionally, the final terms apply elastic-net regularization to the matrices $W$ and $H$ to promote sparsity and mitigate overfitting concerns.

Inspired by iterative optimization processes developed for NMF and SEM (Lee & Seung, 2000; Galeano & Paccanaro, 2022), we develop a multiplicative update rule to minimize the function in Equation 4:

$$W_{i,j} \leftarrow W_{i,j} \times \frac{[XH' + \lambda_{se}XW]_{i,j}}{[WHH' + \lambda_{se}(T \circ (WW'))XX' + \lambda_2 W + \lambda_1 \text{sgn}(W)]_{i,j}} \tag{5}$$

$$H_{i,j} \leftarrow Hi, j \times \frac{[W'X]_{i,j}}{[W'WH + \lambda_2 H + \lambda_1 \text{sgn}(H)]_{i,j}} \tag{6}$$

By initializing $W$ and $H$ with non-negative values, these multiplicative update rules ensure that the elements of the signatures will stay non-negative after each iteration. This iterative process can also be viewed as a gradient descent implementation with an adaptive learning rate:

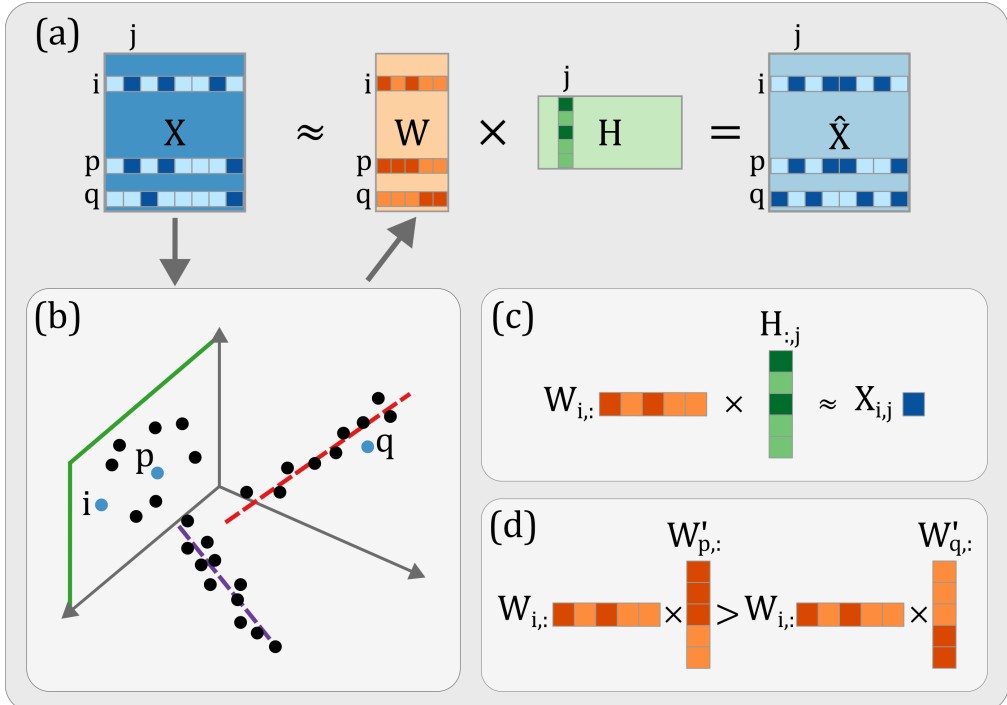

Figure 1: SMF overview. (a) SMF decomposes the association data $X$ into two non-negative matrices $W$ and $H$. By multiplying the matrices $W$ and $H$, we obtain $\hat{X}$, which models $X$, and where all the interactions that were present in $X$ are now replaced by real numbers constituting our predicted scores. (b) Position of the high-dimensional rows of $X$ in the space, SMF uses this information to constrain $W$. (c) The SMF model, like the NMF model, constrains the dot product of the low dimensional representations of a pair of objects $i$ and $j$ to approximate the correspondent element $X_{i,j}$. (d) The dot product of the signatures of two objects $i$ and $p$ that belong to the same subspace in the original association data is enforced to be higher than the dot product of the two objects $i$ and $q$ that does not belong to the same subspace.

$$\alpha_{i,j}^{(W)} = \frac{W_{i,j}}{[WHH' + \lambda_{se}(T \circ (WW'))XX' + \lambda_2 W + \lambda_1 \mathrm{sgn}(W)]_{i,j}},$$

$$\alpha_{ij}^{(H)} = \frac{H_{i,j}}{[W'WH + \lambda_2 H + \lambda_1 \mathrm{sgn}(H)]_{i,j}}.$$

where, in the experiments performed for this work, the loss function tends to decrease after each iteration until the values for $W$ and $H$ meet certain convergence criteria.

After the optimization of the cost function proposed in Equation 4, the obtained signatures are presented as rows in $W$ and columns in $H$. For predicting new associations, the predictions are encoded on the reconstructed matrix $\hat{X} = WH$ as suggested by Equation 3. This approach imposes the SEM constraint only on the rows of $X$. Depending on the problem, it is necessary to choose which elements (the ones represented by columns or the ones represented by rows) are more relevant for the task. In contrast to SEM, which outputs a matrix of distances without providing the distributed representations behind them, SMF manages to obtain those representations by adapting the NMF signatures during the learning to be aware of the subspaces with the matrix $WW'$.

## 4 EXPERIMENTAL RESULTS

### 4.1 DATASETS

**Movielens:** This dataset describes 5-star rating that users give to movies. Group lens makes available a smaller version of the total information for educational and development purposes (Harper & Konstan, 2015). The one used in this work consist in $n_r = 943$ users and $m_r = 1682$ movies, encoded on a matrix $A \in \mathbb{R}^{n_r \times m_r}$. The matrix $X$ contains 10000 non-zero elements representing the known ratings, resulting in an association data matrix with a density equal to 6.3%.

Each movie in $X$ is also associated with a set of $n_g = 18$ genres, these associations are encoded on a matrix $FP \in \mathbb{R}^{m_r \times n_g}$, where if genre $j$ is associated with movie $i$ then $FP_{i,j} = 1$, and is zero otherwise. Different movies and genres are represented by rows and columns respectively, we filtered $FP$ to contain just movies associated with only one genre, resulting in a matrix $FP_{1gen} \in \mathbb{R}^{m_{1gen} \times n_g}$ with $m_{1gen} = 943$. Each user is also associated with their respective gender.

**Frequencies of Drug Side Effects:** Galeano et al. produced a data-matrix containing the frequencies in which certain drugs produce specific side effects (Galeano et al., 2020) by filtering the frequencies obtained from the Side Effect Resource Database (SIDER) (Kuhn et al., 2016). The frequencies are contained on a matrix $R \in \mathbb{R}^{n_r \times m_r}$, with $n_r = 759$ rows representing the drugs and $m_r = 994$ columns representing the side effects. Clinical activity for drugs can be defined using their main Anatomical, Therapeutic and Chemical (ATC) class level, a hierarchical organization of terms where a lower level indicates a more specific descriptor of clinical activity.

### 4.2 MODEL SELECTION AND EVALUATION

As a baseline for evaluating the signatures learned by SMF from the association data, we conduct matrix completion and link prediction tasks. Subsequently, we compare the performance of SMF with that of a regularized version of NMF and SEM. The training process for all three models revolves around minimizing the loss functions outlined in Equations 4, 2, and 1 (with modifications to account for the uncertainty stemming from unknown matrix elements—refer to Section A.1 for more details). Equation 4 is optimized by iteratively updating the values of $W$ and $H$ using the multiplicative update rules as described in 5 and 6. Similarly, NMF and SEM employ their respective multiplicative update rules for learning. The iterative process halts when the parameters being learned satisfy a convergence criterion, where the maximum relative change between consecutive iterations is less than or equal to $1e-3$. Once the training process is over, we obtain the predictions of the models from their respective reconstructed matrices $\hat{X} = WH$ for NMF and SMF, and $\hat{X} = MX$ for SEM. With these predictions, we evaluate the performance of the model for different tasks.

For hyperparameter tuning and evaluation of the models, the datasets are separated into a train, test and validation set each, with 80%, 10%, and 10% of the known associations respectively. Firstly, we train the models only with the train set to evaluate their predictions with the validation set for hyperparameter tuning (see more details in Section A.2). Then, we train the models with the interactions in training and validation in 30 different runs, the reported results are an evaluation of these final predictions compared with the test set.

Root Mean Square Error (RMSE) is used to evaluate the reconstruction, is a metric that quantifies the difference between the values predicted by a model and the observed values, here is measured by comparing the reconstructed entries of the matrices with the known elements on the test set. A low RMSE implies that the scores produced are similar to the true values in the test set. Pearson correlation is also used as a complementary metric, indicating if the predicted scores are ranked in a similar fashion as they are ranked in the test set. A high correlation is desired because it shows if the model is able to closely align with the actual outcomes. The outcomes of the evaluations are reported in Table 3, where the predicted scores of SMF seem to remain closer to the true values than the predicted scores of the competitors.

Another important aspect for SMF is to measure its capabilities in recovering missing associations in the data (not observed links in the graph). For our datasets, this amounts to predicting which movies is a user more likely to watch, and which side effects is a drug likely to cause. With the purpose of tackling this task, we generate two datasets: $R^{int}$ and $X^{int}$. Both of these new datasets were built

Table 1: Results for Movielens dataset

| MODELS | RMSE | CORRELATION | AUROC | AUPRC |
|--------|------|-------------|-------|-------|
| **NMF** | $0.9777 \pm 3\mathrm{e}{-5}$ | $0.5432 \pm 3{-5}$ | $\mathbf{0.9441 \pm 1e{-7}}$ | $0.1402 \pm 5\mathrm{e}{-6}$ |
| **SEM** | $2.9480 \pm 3\mathrm{e}{-6}$ | $0.3531 \pm 3{-7}$ | $0.9436 \pm 4\mathrm{e}{-9}$ | $\mathbf{0.1457 \pm 4{-8}}$ |
| **SMF** | $\mathbf{0.9352 \pm 1e{-5}}$ | $\mathbf{0.5714 \pm 6{-6}}$ | $0.9436 \pm 2\mathrm{e}{-7}$ | $0.1387 \pm 5{-6}$ |

Table 2: Results for SIDER dataset

| MODELS | RMSE | CORRELATION | AUROC | AUPRC |
|--------|------|-------------|-------|-------|
| **NMF** | $0.6558 \pm 1\mathrm{e}{-4}$ | $0.7406 \pm 7\mathrm{e}{-5}$ | $0.8819 \pm 4\mathrm{e}{-10}$ | $0.0879 \pm 4\mathrm{e}{-10}$ |
| **SEM** | $1.8622 \pm 5\mathrm{e}{-6}$ | $0.4181 \pm 4\mathrm{e}{-7}$ | $\mathbf{0.9268 \pm 6e{-8}}$ | $\mathbf{0.1582 \pm 7e{-8}}$ |
| **SMF** | $\mathbf{0.6455 \pm 5e{-5}}$ | $\mathbf{0.7432 \pm 1e{-5}}$ | $0.8582 \pm 2\mathrm{e}{-8}$ | $0.1016 \pm 4\mathrm{e}{-9}$ |

by replacing the known elements on $R$ and $X$ with 1. $R^{int}$ now indicates which movies a user has watched and $X^{int}$ shows which side effect is a drug known to cause. For this binary classification task, the performance is measured using the Area under the receiver operating characteristic curve (AUROC) and the Area under the precision-recall curve (AUPRC). Both RMSE and AUPRC were used for the hyperparameter tuning. The outcomes of the evaluations are reported in Table 3, where SEM outperforms both SMF and NMF in almost every instance, but when one considers the overall performance of the methods SMF ends up as a safer option for a general matrix completion task.

For many link prediction applications, often just a limited number of the most likely links are proposed as new links. This is usually referred to as a top-K recommendation task, where the goal is to find a few specific links that are more likely to exist (Cremonesi et al., 2010). For this purpose, we rank the scores to retrieve the $K$ higher elements within the reconstructed data matrix $\hat{X}$, these would be predicted as new links in the graph, and then we compare them with the test set to obtain the precision at top-K, that is, the ratio of known positives within the links predicted as positives. To have fair measurements, the true positives for the analysis are the ones on the test set, and all the unknown elements on the original datasets are considered as true negatives (Krichene & Rendle, 2020). The outcomes of the evaluations are reported in Figure 2, where SMF clearly outperforms the competitors in almost every setting.

### 4.3 SIGNATURES ANALYSIS

To further demonstrate that the SMF-derived signatures offer a more meaningful encoding of previously unseen latent processes, we will analyze these low-dimensional representations and compare them with the ones learned by NMF. Our aim is to assess their capacity for encapsulating inherent data characteristics, which were not part of the training process but may play crucial roles in establishing connections between objects. The objective of this study is twofold: firstly, to verify if SMF effectively clusters objects into meaningful groups within the low-dimensional space; and secondly, to determine whether SMF achieves superior class separation of objects compared to NMF.

We begin by grouping the signatures in $W$ into disjoint sets based on the respective classes of the objects they represent. Subsequently, we calculate a similarity matrix, $W_{sim} \in \mathbb{R}^{n \times n}$, containing the cosine similarity between all the signatures. Finally, we employ a two-sample $t$-test to assess whether the similarities between objects within the same class differ significantly from the similarities between objects in different classes. An illustration of this procedure is provided in Figure 3a. This process was repeated 30 times across different runs of the models.

For the Movielens dataset, we organized users based on their gender, while movies were grouped by their respective genres. In our genre analysis, we refined the dataset to include only movies with a single genre, enabling the classification of movies into distinct categories. In the case of the SIDER dataset, we categorized drugs according to their various levels within the ATC hierarchy[1]. The lower

---

[1] ATC categories were obtained from the ATC codes WHO 2018 release.

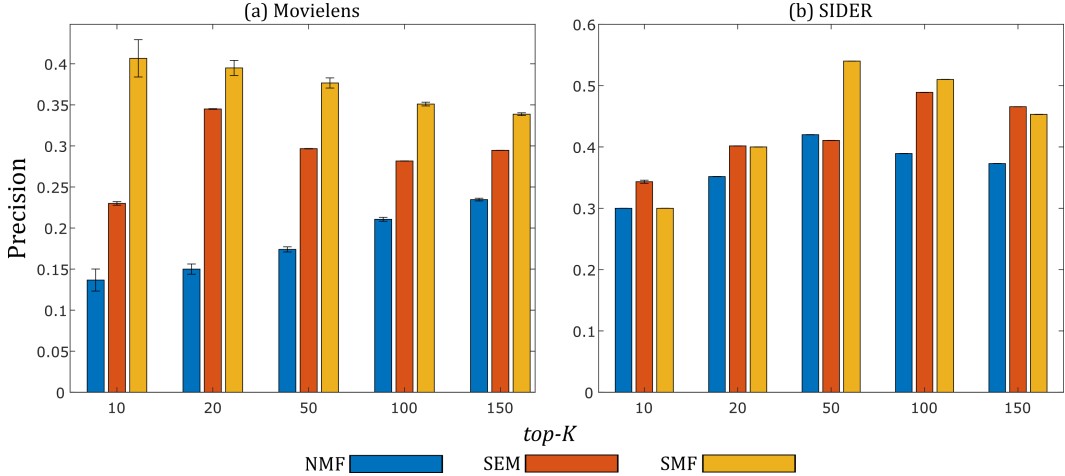

Figure 2: Precision at top-K: Bar plot of the precision of NMF, SEM and SMF for different values of $K$ while predicting missing links in the interaction data. The error bars indicate the variance of the precision for 30 different runs of the models. (a) Precision for the Movielens dataset while predicting links between users and movies, the negative to positive ratio in the test set is approximately 1600. (b) Precision for the SIDER dataset while predicting links between side effects and drugs, the negative to positive ratio in the test set is approximately 200.

levels of the ATC hierarchy provide more specific terms for drug classification. At the first level, drugs fall into one of 14 classes, and at subsequent levels, drugs are associated with increasingly specific terms, ensuring that they are only grouped with drugs already classified at earlier levels. For this study, we compared the similarity of drug signatures across three levels of the hierarchy: anatomical, therapeutic, and pharmacological.

**SMF consistently clusters objects in the low-dimensional space.** In the analysis of the Movielens-gender dataset, NMF achieves statistical significance in 76% of the runs, while SMF consistently attains significant separation in all runs (100%). When examining movie genres, these percentages shift to 100% for NMF and 93% for SMF. However, when considering the various levels of the ATC hierarchy, NMF struggles to maintain consistent separation of drugs into classes, achieving statistical significance in only 3%, 13%, and 13% of runs for the $1^{st}$, $2^{nd}$, and $3^{rd}$ levels, respectively, while SMF reaches 100% across all the levels. The consistent achievement of statistical significance in this experiment indicates effective clustering of objects in the low-dimensional space. It also serves as compelling evidence that the signatures reliably encode meaningful information pertaining to fundamental attributes of the objects.

**SMF achieves superior class separation.** To assess the efficacy of each method in achieving class separation, we employ the Z-score difference between the means of the intra-class and inter-class similarity distributions. This Z-score difference provides an effective measure of the actual distinction between these distributions, leveraging both mean and variance to compute a form of distance. A larger Z-score signifies greater separation between the distributions.

Our results for this experiment are summarized in Figures 3b and 3c. Overall, it becomes evident that SMF-learned signatures effectively group objects into more meaningful clusters than those learned by NMF, across both datasets and diverse groups. Notably, in Figure 3c, we observe that the separation between groups in the ATC levels increases as we delve deeper into the hierarchy. This reflects the fact that the drug clinical activity becomes more similar as we move to more specific levels.

## 5 DISCUSSION & CONCLUSION

In this work, we introduced Self-Matrix Decomposition (SMF), a constrained NMF approach that learns low-dimensional representations while aiming to preserve the subspace information present in the association data. Our objective is to encode these subspaces by enforcing a higher dot product

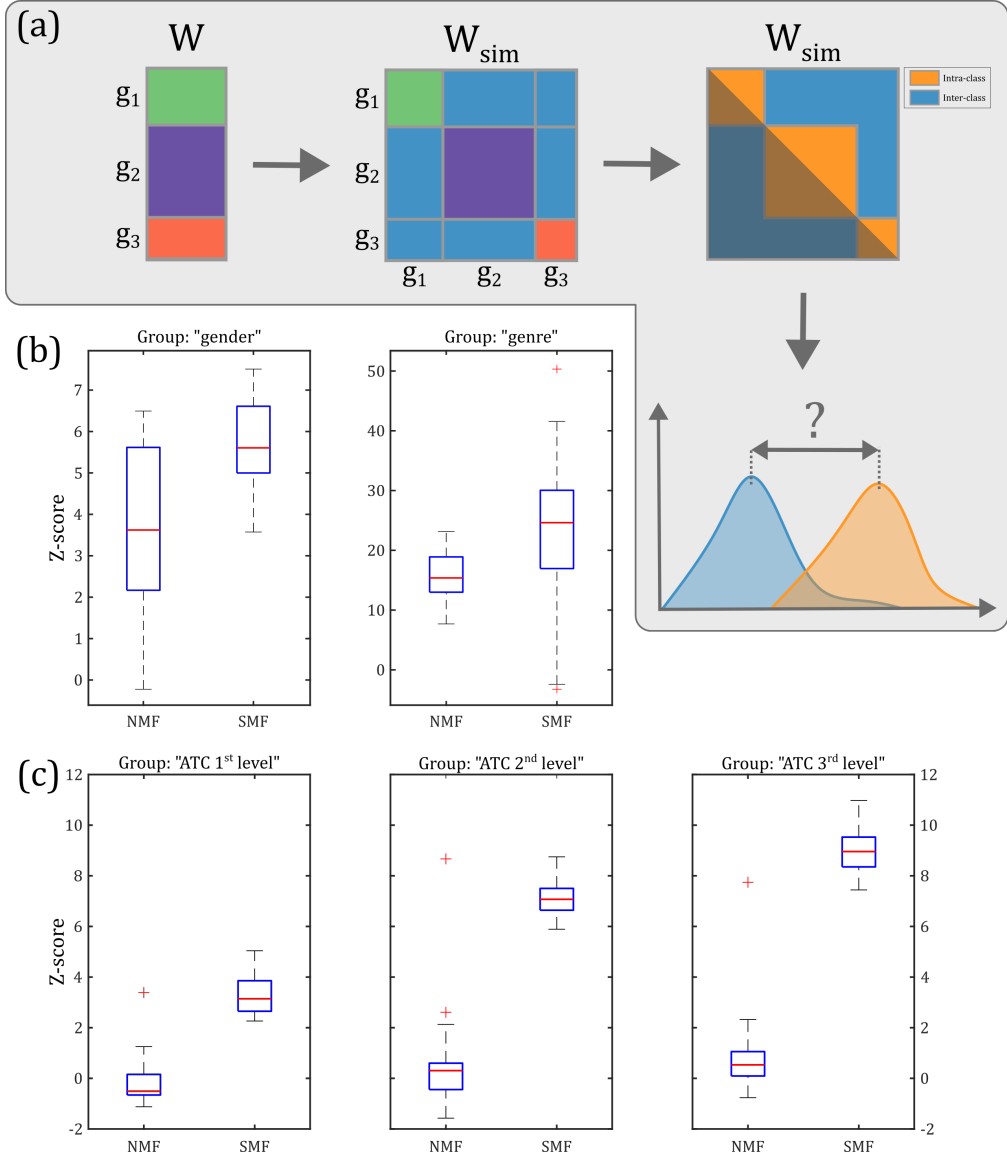

Figure 3: Signature Analysis. (a) Pipeline explaining the experiment, first, we have the signature matrix $W$, arranged into three different groups, $g_1$, $g_2$, and $g_3$. Next, $W_{sim}$ contains the similarities between all the signatures. Lastly, we calculate if there is a statistical significance in the difference between the intra-class and inter-class similarities from $W_{sim}$ (b) Movielens experiments. Box plots showing the distributions of the Z-score differences across 30 different runs of NMF and SMF. The left plot shows the separation between the distributions of the similarities while grouping the users by their gender (Male and Female for this dataset). The right plot shows the separation between the distribution of the similarities while grouping the movies by their genres (18 distinct groups). (c) ATC-category experiments. Box plots showing the distributions of the Z-score differences across 30 different runs of NMF and SMF. From left to right, the grouping advances in the ATC hierarchy, $1^{st}$: anatomical, 2.

between the signatures of two objects localized within the same sub-space in the association data matrix than the dot product of the signatures of two objects residing in different sub-spaces. This constraint also aids in restricting the solution space for the learned matrices $W$ and $H$, mitigating the non-uniqueness issue of NMF. It is important to note that once a proper solution for $W$ and $H$ is learned, obtaining equivalent solutions $W^* = WD^{-1}$ and $H^* = DH$ is not recommended. While

these solutions can reconstruct the same matrix $\hat{X}$ they fail to reflect the subspace information. This happens because the new matrices $W^*$ and $H^*$ exhibit a different relationship between the components of the signatures compared to $W$ and $H$.

SMF can decompose low-rank matrices while preserving their subspaces in $WW'$, leveraging the process by which $M$ is learned in Equation 2. The two dot product constraints in SMF (as illustrated in Figures 1c and 1d) facilitate the signatures to learn better representations and capture the latent processes, as they allow the signatures to glean information from objects residing within the same subspace (as depicted in Figure 1b).

The experimental results strongly indicate that SMF exhibits comparable or superior predictive capabilities compared to both NMF and SEM. This indicates that the proposed constraint in SMF can effectively extract valuable insights from known interactions. SMF attains better RMSE values compared to NMF, suggesting that the subspaces encoded in the matrix coefficient of SEM contribute to the learning of more descriptive signatures for recommendations. Although AUROC and AUPRC are typically employed to evaluate classification tasks, they primarily reveal how true positive samples are ranked when predicting the likelihood of an association. However, in practical terms, the effectiveness of a method in suggesting new associations, or the proximity of true positives to the top of the ranking, is of greater interest. This is quantified by the precision at the top-K metric, where SMF consistently outperforms NMF and SEM in nearly every instance. This reinforces the assertion that SMF signatures are adept at capturing the distinctive patterns typically learned by NMF and SEM from association data.

We conducted experiments to assess whether a set of known object properties could be effectively encoded within the object signatures. Prior research has also delved into similar investigations, revealing, for instance, that various NMF variants can learn signatures encoding movie genres (Koren et al., 2009) and drug ATC categories (Galeano et al., 2020). Subsequently, we conducted an in-depth analysis of the similarities among signatures generated by NMF and SMF across multiple runs and diverse groupings. This analysis aimed to ascertain whether objects belonging to the same group consistently clustered together in the low-dimensional space. The experimental results provide compelling evidence, clearly demonstrating that SMF consistently achieves superior class separation in all conducted experiments.

Furthermore, it becomes apparent that SMF offers significantly higher stability in learning signatures compared to NMF. This is evident from the fact that in multiple runs, SMF achieves statistical significance approximately $99\%$ of the time, whereas NMF accomplishes this feat in only $41\%$ of the cases. This finding reinforces our earlier assertion regarding the challenges posed by NMF in selecting high-quality signatures from countless of potential solutions for the same problem. In contrast, SMF operates within a more constrained solution space, thanks to the SEM term in Equation 4, consistently yielding meaningful object representations.

### AUTHOR CONTRIBUTIONS

If you'd like to, you may include a section for author contributions as is done in many journals. This is optional and at the discretion of the authors.

### ACKNOWLEDGMENTS

Use unnumbered third level headings for the acknowledgments. All acknowledgments, including those to funding agencies, go at the end of the paper.

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

## A  APPENDIX

### A.1  LEARNING OF THE MODELS

To train the models for the results reported in Section 4, the Equations 4, 1 and 2 were adapted in order to address the impact that unknown interactions may have on the learning, assigning more importance to the reconstruction of known elements. Similarly as in data-driven regularized NMF (Galeano et al., 2020), we can define a value $\alpha$ that modulates the importance of the zeros in $X$ during the learning. For this purpose, we define a matrix $P \in \mathbb{R}^{n \times m}$ (Blondel et al., 2008), where $P_{i,j} = \alpha$ if $X_{i,j} = 0$ and $P_{i,j} = 1$ otherwise, the new weighted SMF loss function is:

$$\min_{W,H} \mathcal{L}_{\text{WSMF}}(W,H) = \frac{1}{2}\|P \circ (X - WH)\|_F^2 + \frac{\lambda_{se}}{4}\|P \circ (X - [T \circ (WW')]X)\|_F^2$$
$$+ \lambda_1\|W\|_1 + \lambda_1\|H\|_1 + \frac{\lambda_2}{2}\|W\|_F^2 + \frac{\lambda_2}{2}\|H\|_F^2 \tag{7}$$
$$\text{subject to } W, H \geq 0.$$

We developed the following multiplicative update rules to optimize this version of the loss function of SMF:

$$W_{i,j} \leftarrow W_{i,j} \times \frac{[(P^2 \circ X)H' + \lambda_{se}(P^2 \circ X)W]_{i,j}}{[(P^2 \circ (WH))H' + \lambda_{se}(P^2 \circ ((T \circ (WW'))X))X' + \lambda_2 W + \lambda_1\text{sgn}(W)]_{i,j}} \tag{8}$$

$$H_{i,j} \leftarrow Hi, j \times \frac{[W'(P^2 \circ X)]_{i,j}}{[W'(P^2(WH)) + \lambda_2 H + \lambda_1\text{sgn}(H)]_{i,j}} \tag{9}$$

**NMF**: Here we opted to train a Regularized NMF (Pauca et al., 2006), where we modify Equation 1 by regularizing $W$ and $H$ via elastic-net (Zou & Hastie, 2005) as shown in the following loss function:

$$\min_{W,H} \mathcal{L}_{\text{WNMF}}(W,H) = \frac{1}{2}\|P \circ (X - WH)\|_F^2 \tag{10}$$
$$\text{subject to } W, H \geq 0.$$

that can be optimized with the following multiplicative update rules:

$$W_{i,j} \leftarrow W_{i,j} \times \frac{[(P^2 \circ X)H']_{i,j}}{[(P^2 \circ (WH))H' + \lambda_2 W + \lambda_1\text{sgn}(W)]_{ij}} \tag{11}$$

$$H_{i,j} \leftarrow Hi, j \times \frac{[W'(P^2 \circ X)]_{ij}}{[W'(P^2(WH)) + \lambda_2 H + \lambda_1\text{sgn}(H)]_{ij}} \tag{12}$$

**SEM**: here we modified Equation 2 by replacing the diagonal constraint by adding the trace of $M$ multiplied by a large number $\gamma$ as a new term in the loss function (Galeano & Paccanaro, 2022):

$$\min_{M} \mathcal{L}_{\text{WSEM}}(M) = \frac{1}{2}\|P \circ (X - MX)\|_F^2 + \frac{\lambda_2}{2}\|M\|_F^2 + \lambda_1\|M\|_1 \tag{13}$$
$$\text{subject to } M \geq 0$$

that can be optimized with the following multiplicative update rules:

$$M_{i,j} \leftarrow M_{i,j} \frac{[(P^2 \circ X)X]_{i,j}}{[(P^2(MX))X' + \gamma I + \lambda_2 M + \lambda_1\text{sgn}(M)]_{i,j}} \tag{14}$$

It can be proven that the optimization proposed for Equation 10 and 13, both loss functions are non-increasing at their respective parameters converge in a local minimum for NMF, and global minimum for SEM. For Equation 7, we observed that the loss function always increases for the first iteration, then, for the second iteration forward, the loss function turns out to be non-increasing, and it also manages to achieve convergence for every run of the model, what seems to indicate that this version of adaptive gradient descent is well suited for the problem under study in this work. The algorithm for the optimization of Equation 7 was implemented in Matlab R2023a, and the code is

included with this submission. The training is stopped by satisfying the stopping criteria $\delta \leq 1e-3$ for all the trained models, and the maximum relative change $\delta$ is defined as:

$$\delta = \frac{\max(\|W_{i,j}^{\text{old}} - W_{i,j}^{\text{new}}\|)}{\max(\|W_{i,j}^{\text{old}}\|)} \tag{15}$$

where $W^{\text{old}}$ and $W^{\text{new}}$ are the values of the matrix $W$ after each iteration, clearly the same formula is also applied for $H$ and $M$.

## A.2 HYPERPARAMETER TUNING

Hyperparameter tuning was performed as explained in Section 4.2, first we choose a value for the rank of the factorization $k$ for NMF and SMF without regularization and $\alpha = 0$. Then we perform a greed search to set the values for the regularization weights and the importance of the zeros $\alpha$ for all the methods for the two different datasets. The final set of hyperparameters selected to perform the experiments are detailed in Table A.2

Table 3: Results for Movielens dataset

| Values | NMF | SEM | SMF | NMF | SEM | SMF |
|---|---|---|---|---|---|---|
| $k$ | 10 | $(-)$ | 10 | 10 | $(-)$ | 10 |
| $\lambda_1$ | 0.5 | 0.5 | 1 | 0.5 | 0.5 | 0.5 |
| $\lambda_2$ | 0.5 | 0.5 | 0 | 0.5 | 0.5 | 0.5 |
| $\alpha$ | 0.224 | 0.224 | 0.224 | 0.0025 | 0.224 | 0.0025 |

For obvious reasons, SEM does not have a value set for $k$. It is important to note that the value of $\alpha$ in Table A.2 for Movielens is used only for the link prediction task (when evaluating AUROC, AUPRC and precision). For rating prediction (when evaluating RMSE and correlation), $\alpha$ is set to zero, reflecting the fact that there are no true zeros in the dataset, because in the ideal scenario, where all the users assign a rating to all the movies those values should be between $1$ and $5$.

