# OpenReview forum: "Constraining Non-Negative Matrix Factorization to Improve Signature Learning"
_ICLR.cc/2024/Conference — Submitted to ICLR 2024_

### Official Review · Reviewer_G6Gq · 2023-10-13

**Soundness:** 2 fair
**Presentation:** 2 fair
**Contribution:** 2 fair
**Rating:** 3
**Confidence:** 5

**Summary:**

This paper proposed a combination model from NMF and SEM to conduct signature learning, and validated its effectiveness through two experiments w.r.t. matrix completion and link prediction tasks.

Overall, this work looks solid but with very limited novelty.

**Strengths:**

The writting logic is good.

**Weaknesses:**

1. the proposed SMF could be viewed as a combination of NMF and SEM, so its novelty is limited.
2. some mathematical symbols are not clearly stated, such as A and X in section 4.1, and \alpha in section 4.2.
3. the selected competitive baselines can be more richful, such as spareNMF [1], GraphNMF [2].


[1] Patrik O. Hoyer: Non-negative Matrix Factorization with Sparseness Constraints. J. Mach. Learn. Res. 5: 1457-1469 (2004)
[2] Shangming Yang, Zhang Yi, Mao Ye, Xiaofei He: Convergence Analysis of Graph Regularized Non-Negative Matrix Factorization. IEEE Trans. Knowl. Data Eng. 26(9): 2151-2165 (2014)

**Questions:**

I have no questions.

---

> ### Author Response · Authors · 2023-11-22
> **Response**
>
> We appreciate the reviewer's time and insightful comments. Here, we provide a detailed response to each point raised in the review.
>
> **The proposed SMF could be viewed as a combination of NMF and SEM, so its novelty is limited.**
>
> We thank the reviewer for bringing out this point. We will improve the explanation of how our method relates to NMF and SEM in the revised version of the manuscript.
>
> SMF is not a straightforward combination of NMF and SEM. It can be thought of as a new model that relies on NMF and constrains the space of solutions by using ideas inspired by SEM. SEM itself is not directly plugged into the model. We proposed a “self-expressive term” that differs from the original term used by SEM. One of our insights was including the low dimensional representation into the self-expressive term and observing that this results in representations that can preserve better the proximities from original sub-spaces. We then derived the multiplicative rules of our model and showed that SMF can take advantage of both NMF and SEM strategies.
>
>
> **The selected competitive baselines can be more richful, such as spareNMF [1], GraphNMF [2]**
>
>
> We thank the reviewer for this suggestion.
>
> Although we agree that regularizations used by sparseNMF [1] and GraphNMF[2] can help to obtain better representations, we did not select them as competitive baselines for the following reasons:
>
> - Our proposed method, SMF, and our competitive baseline (regularized NMF) already rely on L1 norm and L2 norm regularizations. Because the sparseness constraints used by sparseNMF also rely on L1 and L2 norms, we think that adding it as a competitive baseline would not be very informative for our discussions. We believe that sparseNMF would have a different behavior than the regularized NMF used as our baseline only if one wants to customize the sparsity according to the particularities of a given problem. However, in a general scenario, we believe that the chosen baseline (regularized NMF) already incorporates the main advantages mentioned by [1].
>
> - Our paper focused on association data. Although in theory graphNMF could be applied to decompose any nonnegative matrix, its main goal is finding a parts-based representation space from an input matrix in which each column corresponds to a sample vector. Then it constructs a knn graph based on the distances between the vectors. The applicability of graphNMF was shown in the context of recovering better parts of images.  For an association matrix, it is not clear whether the rows or columns should correspond to nodes in the graph, and which parts should be recovered in the learned representations. Thus, we believe that there is no strong motivation for applying graphNMF to an association matrix.
>
> It is also important to notice that the main idea of SMF is incorporating constraints inspired by SEM into an NMF model. Thus, both sparseNMF and graphNMF could be adapted as well to incorporate these constraints.

---

### Official Review · Reviewer_NTRu · 2023-10-30

**Soundness:** 2 fair
**Presentation:** 2 fair
**Contribution:** 2 fair
**Rating:** 6
**Confidence:** 3

**Summary:**

This paper presents a Non-negative Matrix Factorization (NMF) approach, named Self-Matrix Factorization (SMF), which factorizes association matrix into low dimensional representations. The proposed solution is derived from Self-Expressive Models (SEM). The representations or so-called signatures in the paper are learned by imposing  the reconstruction to use only data points that lie in the same low dimensional subspace. This way can guarantee the solutions are within some confined space. In addition, the paper states that the proposed method is robust and also can capture meaningful relationships. Experimental results showed that the proposed SMF product comparable or better prediction results than other existing NMF or SEM methods.

**Strengths:**

1. The paper is easy to read and understand. The concept of proposed solution is simple.
2. The low dimensional representations are meaningful and the intra-class distance of the factorization is larger compared to results from other methods.
3. Results are produced by real world datasets, and the precision-recall curve is comparable to other methods.

**Weaknesses:**

1. It is somewhat incremental improvement from SEM. The results are comparable, not significantly better.
2. How to interpret the meaningful relationships in the low dimensional representations need more elaboration.

**Questions:**

1. Some metrics are not well explained in the paper. What is the formula to compute Z-score?
2. Are the low dimensional data representations stable? How would it change if small amount of data are added or deleted from association matrix?
3. In figure 2, Precision at top-K decreases in the proposed method, especially for movielens dataset. Is there any explanation?

**Details Of Ethics Concerns:**

No concern

---

> ### Author Response · Authors · 2023-11-13
> **review possibly of a different paper**
>
> We believe that the reviewer has uploaded the review for a different paper.

---

> > ### Comment · Reviewer_NTRu · 2023-11-13
> >
> > The correct review has been updated. Thanks for pointing out the mistake.

---

> ### Author Response · Authors · 2023-11-22
> **Response**
>
> We appreciate the reviewer's time and insightful comments. Here, we provide a detailed response to each point raised in the review.
>
> **It is somewhat incremental improvement from SEM. The results are comparable, not significantly better.**
>
>
> We thank the reviewer for bringing out this point. Although we agree that some results are comparable for recommendation tasks, and we do mention this in the paper, we think that this is not necessarily a weakness, since the main goal of the paper is to show that we can obtain more meaningful representations. Thus, our main results are shown in the Signature Analysis section, whereas the recommendation tasks illustrate that better representations can lead to better or comparable predictive power. Nevertheless, it is interesting to notice that in the overall picture of the recommendation tasks, SMF has a significantly lower RMSE than SEM while also maintaining a higher precision than NMF.
>
>
> **How to interpret the meaningful relationships in the low dimensional representations need more elaboration.**
>
> We thank the reviewer for this suggestion. We will clarify the interpretation analysis and what we mean by “meaningful relationships” in the revised version of the manuscript.
>
> When we say that our model captures meaningful relationships between objects, we mean that the similarity between two signatures reflects similarities in terms of latent processes underlying the data. Since we assume that the latent features of our model encode relevant processes, we expect that the learned signatures of two objects will be similar if they share latent processes, and dissimilar, otherwise. To test this hypothesis, we leverage additional knowledge about the objects, which was not utilized during the learning process. This enables us to cluster objects that are expected to share latent processes. For instance, drugs within the same ATC category are likely to share molecular mechanisms. If our hypothesis holds true, we expect that drugs within the same ATC category will exhibit more similar signatures than those from different categories.
>
>
>
>
> **Some metrics are not well explained in the paper. What is the formula to compute Z-score?**
>
> Thank you for pointing this out. We will include the formula and explanation in the revised version of the manuscript.
>
> We use the Z-score from the two-sample z-test to measure the difference between signature similarities of objects that are from the same group and objects that are from different groups. Given the average signature similarity of objects that are in the same group ($\mu_in$), the average signature similarity of objects that are in different groups ($\mu_out$), and the corresponding standard deviations (\sigma_{in} and \sigma_{out}), the z-score formula is:
>
> $Z = \frac{ \mu_{in} - \mu_{out}}{\sqrt{\frac{\sigma_{in}^2}{n_{in}} + \frac{\sigma_{out}^2}{n_{out}}}}$,
>
> where $n_{in}$ and $n_{out}$ are the number of object pairs intra- and inter-groups, respectively.
>
> We can interpret the z-score as a normalized distance that measures how different two distributions are by adjusting the difference between the means according to their standard deviation.
>
>
> **Are the low dimensional data representations stable? How would it change if small amount of data are added or deleted from association matrix?**
>
> As evidenced by our experiments, the generated signatures consistently yield stable results across different runs. To address the missing data concern, we ran experiments similar to the ones detailed in Section 4.3 for signature analysis, but we specifically trained the model using only 90% of the known interactions in the SIDER dataset. Notably, even with this reduced dataset, the difference in results compared to using the full matrices is minimal.
>
> Here we reported the means of the Z-Score difference between the means for both models NMF and SMF:
>
> | Models | ATC $1^{st} Level$ | ATC $2^{nd} Level$ | ATC $3^{rd} Level$ |
> | --------- |  :--------------: | :--------------: | :--------------: |
> | NMF | $-0.1013$ | $0.3934$ | $0.7411$ |
> | SMF | $2.9746$ | $7.0157$ | $8.5293$ |
>
> We plan to run similar experiments for the other datasets and include them in the revised version of the manuscript.
>
>
> **In figure 2, Precision at top-K decreases in the proposed method, especially for movielens dataset. Is there any explanation?**
>
> For that particular case, the first top-K recommendations already contain a high amount of true positives, which results in high precision. As the value of K increases, the amount of true positives does not increase accordingly, as most of them were already captured in the highest value for K. Note that the same trend can be observed for SEM when examining the precision score for K equal to 20 and upwards.

---

### Official Review · Reviewer_oagP · 2023-11-01

**Soundness:** 2 fair
**Presentation:** 3 good
**Contribution:** 1 poor
**Rating:** 3
**Confidence:** 4

**Summary:**

This paper revisits the Nonnegative Matrix Factorization problem, by adding extra constraints inspired by the Self Expressive Models. The authors present some empirical results on different accuracy and qualitative metrics.

**Strengths:**

The paper is well-written and has a good empirical analysis of the chosen datasets.

**Weaknesses:**

The area of NMF is very well studied, and the truth is that with more modern methods inspired by relational learning, GNNs can do a much better job at the applications in which NMF has been used. The paper has some good intuitions, and the idea of introducing the Self Expressiveness on the W matrix subspace is interesting. However, we don't have any theoretical guarantees that this is going to work in general. I would accept intuition as a guidance, but:
- NMF suffers from local minima, and this needs to be studied extensively, even empirically
- The number of experiments is too small to get some significant guarantees. We need a much bigger pool of benchmarks
- Scalability is another issue not examined. This is strongly tied to runtimes and convergence steps needed

**Questions:**

Can the authors quantify the robustness to local minima?

---

> ### Author Response · Authors · 2023-11-22
> **Response(1/2)**
>
> We appreciate the reviewer's time and insightful comments. Here, we provide a detailed response to each point raised in the review.
>
> **The area of NMF is very well studied, and the truth is that with more modern methods inspired by relational learning, GNNs can do a much better job at the applications in which NMF has been used.**
>
> We acknowledge the reviewer's point regarding NMF being well studied and that there are different methods that can achieve a better recommendation performance. However, it’s important to note that NMF remains a prevalent and impactful technique within the field, and holds the potential to outperform even deep learning-based architectures [1] while obtaining meaningful representations for objects.
> In line with the conference's focus on learning representations, our paper aims to enhance the learning of the meaningful representations that you would normally obtain from NMF. These are learned from complementary analysis strategies and extracting patterns from observed association data. This aspect holds significance as SMF has the ability to provide insights into aspects of the problem, such as identifying categories of drugs that were not explicitly involved in the training process.
> In the paper, we report recommendation metrics only as indirect evidence of the capabilities of SMF in the context of recommendation tasks. We don’t claim that this version of matrix decomposition is an effective state-of-the-art recommender. While graph neural networks (GNN) and other deep learning strategies are frequently employed in various domains, including recommendation systems, their interpretability often lags behind. In contrast, tasks such as interpretability are more intuitive for NMF and SEM.
> We intend to revise the manuscript by incorporating the relevant citation and providing a more detailed explanation to elucidate this particular point.
>
> [1] Rendle, Steffen, et al. "Neural collaborative filtering vs. matrix factorization revisited." Proceedings of the 14th ACM Conference on Recommender Systems. 2020.
>
> **NMF suffers from local minima, and this needs to be studied extensively, even empirically.**
>
> We acknowledge that the optimization process described in this paper suffers from local minima, similar to many other machine learning frameworks in the literature, and we do not believe this is a significant problem for the task of learning meaningful representations. Our reported results obtained from different runs of our framework with different initializations were shown to be stable, indicating that SMF is able to consistently propose similar solutions. We will modify the paper to clarify that the proposed optimization process does not offer any robustness to local minima.
>
> **The number of experiments is too small to get some significant guarantees. We need a much bigger pool of benchmarks.**
>
> To address this concern, we selected an additional dataset, called ModCloth [2], where users assign ratings to clothes. Due to the time constraints in the rebuttal phase, we conducted experiments only on one additional dataset. We also had to skip the exhaustive hyperparameter tuning and perform a subsetting of the dataset to manage computational time. In the revised version of the manuscript, we intend to include the full set of experiments on the complete or a less subsetted, version of this dataset.
>
> After randomly subsetting the data, we obtain a data matrix $C \in \mathbb{R}^{n_c \times m_c}$ for $ n_c = 5419$ clothing items and $m_c = 32.089$ users with a density of $\approx 0.05 \%$. Each clothing item belongs to $1$ of $66$ different categories that are used for the signature analysis. The results shown in the table below correspond to one run of NMF, SEM and SMF. Similarly to the other datasets, SMF clearly outperforms the other methods and achieves better category separation in contrast to NMF.
>
>
>
> | Models | RMSE    | Correlation | AUROC  | AUPRC   | Z-score |
> | ------ | :-----: | :---------: | :----: | :-----: | :-----: |
> | NMF    | $1.5847$ | $0.0801$ | $0.6871$ |$1.64\mathrm{e}{-4}$ | $62.34$ |
> | SEM    | $4.3112$ | $0.0423$ | $0.6230$ |$2.50\mathrm{e}{-4}$ | (-) |
> | SMF    | $1.3291$ | $0.1012$ | $0.7515$ |$2.84\mathrm{e}{-4}$ | $238.35$ |
>
>
> [2] Misra, Rishabh, Mengting Wan, and Julian McAuley. "Decomposing fit semantics for product size recommendation in metric spaces." Proceedings of the 12th ACM Conference on Recommender Systems. 2018.

---

> > ### Author Response · Authors · 2023-11-22
> > **Response (2/2)**
> >
> > **Scalability is another issue not examined. This is strongly tied to runtimes and convergence steps needed.**
> >
> > Here we address the issue of the scalability of our method, in the following table illustrates the mean time of each iteration and the mean number of steps necessary for convergence for the Movielens and the Cloth datasets. We will modify the paper accordingly to report these characteristics.
> >
> > **Time per iteration in seconds:**
> >
> > | Models | Movielens    | ModCloth |
> > | ------ | :----------: | :---: |
> > | NMF    | $0.01531$      | $1.4287$|
> > | SEM    | $0.04338$      | $7.3948$|
> > | SMF    | $0.02797$      | $7.4693$|
> >
> > **Iterations for convergence:**
> >
> > | Models | Movielens    | ModCloth |
> > | ------ | :----------: | :---: |
> > | NMF    | $1868.20$      | $467$   |
> > | SEM    | $244.20$       | $292$   |
> > | SMF    | $1427.733$     | $1032$  |

---

### Meta-Review · Area_Chair_mLXK · 2023-12-04

**Metareview:**

The reviewers were generally not convinced by the scalability, theoretical guarantees, novelty (the method being a combination of NMF and SEM), sufficiency of empirically evaluations, and broad applicability of the proposed method. At present, there are many issues that require the authors to do a complete rewrite for the paper to be accepted at a top-tier ML conference.

**Justification For Why Not Higher Score:**

There are several weaknesses as highlighted above.

**Justification For Why Not Lower Score:**

The score is already rather low.

---

### Decision · Program_Chairs · 2024-01-16

Reject